# Screening of the Drug-Induced Effects of Prostaglandin EP2 and FP Agonists on 3D Cultures of Dexamethasone-Treated Human Trabecular Meshwork Cells

**DOI:** 10.3390/biomedicines9080930

**Published:** 2021-07-31

**Authors:** Megumi Watanabe, Yosuke Ida, Masato Furuhashi, Yuri Tsugeno, Hiroshi Ohguro, Fumihito Hikage

**Affiliations:** 1Departments of Ophthalmology, School of Medicine, Sapporo Medical University, Sapporo 060-8556, Japan; watanabe@sapmed.ac.jp (M.W.); funky.sonic@gmail.com (Y.I.); yuri.tsugeno@gmail.com (Y.T.); ooguro@sapmed.ac.jp (H.O.); 2Departments of Cardiovascular, Renal and Metabolic Medicine, Sapporo Medical University, Sapporo 060-8556, Japan; furuhasi@sapmed.ac.jp

**Keywords:** 3D spheroid culture, human trabecular meshwork, omidenepag isopropyl, omidenepag, PGF2α, dexamethasone

## Abstract

The objective of the current study was to perform a screening of the drug-induced effects of the prostaglandin F2α (PGF2α) and EP2 agonist, omidenepag (OMD), using two- and three-dimensional (2D and 3D) cultures of dexamethasone (DEX)-treated human trabecular meshwork (HTM) cells. The drug-induced effects on 2D monolayers were characterized by measuring the transendothelial electrical resistance (TEER) and fluorescein isothiocyanate (FITC)–dextran permeability, the physical properties of 3D spheroids, and the gene expression of extracellular matrix (ECM) molecules, including collagen (COL) 1, 4 and 6, and fibronectin (FN), α smooth muscle actin (αSMA), a tissue inhibitor of metalloproteinase (TIMP) 1–4, matrix metalloproteinase (MMP) 2, 9 and 14 and endoplasmic reticulum (ER) stress-related factors. DEX induced a significant increase in TEER values and a decrease in FITC–dextran permeability, respectively, in the 2D HTM monolayers, and these effects were substantially inhibited by PGF2α and OMD. Similarly, DEX also caused decreased sizes and an increased stiffness in the 3D HTM spheroids, but PGF2α or OMD had no effects on the stiffness of the spheroids. Upon exposure to DEX, the following changes were observed: the upregulation of *COL4* (2D), α*SMA* (2D), and *TIMP4* (2D and 3D) and the downregulation of *TIMP1* and *2* (3D), *MMP2* and *14* (3D), inositol-requiring enzyme 1 (IRE1), activating transcription factor 6 (ATF6) (2D), and glucose regulator protein (GRP)78 (3D). In the presence of PGF2α or OMD, the downregulation of *COL4* (2D), *FN* (3D), *α**SMA* (2D), *TIMP3* (3D), *MMP9* (3D) and the CCAAT/enhancer-binding protein homologous protein (CHOP) (2D), and the upregulation of *TIMP4* (2D and 3D), MMP2, 9 and 14 (2D), respectively, were observed. The findings presented herein suggest that 2D and 3D cell cultures can be useful in screening for the drug-induced effects of PGF2α and OMD toward DEX-treated HTM cells.

## 1. Introduction

It is well recognized that glaucomatous optic neuropathy (GON), a progressive and chronic optic neuropathy caused by elevated intraocular pressures (IOPs), ultimately leads to the development of irreversible blindness [1,2,3]. The mechanism responsible for the IOP elevation has been proposed to involve an increase in the resistance to the outflow of aqueous media from the trabecular meshwork (TM), presumably caused by the deposition of higher than normal levels of extracellular matrix (ECM) molecules such as collagen molecules (COLs), fibronectin (FN), and others leading to the development of both primary open-angle glaucoma (POAG) and steroid-induced glaucoma (SG) [4,5]. Thus, in addition to the administration of anti-glaucoma medication as the initial therapy, laser treatment or surgical intervention are sometimes required for the treatment of GON [6].

Prostaglandin (PG) F2α agonists (PGF2α-ags) have been extensively used as first-line drugs for the treatment of GON, based on their powerful hypotensive effect through their FP receptors, in addition to the fact that they have few serious systemic side effects [7,8,9]. Quite recently, in addition to PGF2α-ags, a selective, nonprostaglandin, prostanoid EP2 agonist, omidenepag isopropyl (OMDI), which is hydrolyzed in vivo into the active form (omidenepag, OMD), has recently been made available for the treatment of patients with ocular hypertension (OH) and GON [10,11]. In terms of the mechanisms responsible for causing these hypotensive effects, it is generally thought that both PGF2α-ags and the EP2 agonist OMD increase the ease of uveoscleral outflow, but not the ease of conventional TM outflow [12]. Interestingly, a study using OH monkeys reported that the pharmacokinetics of aqueous outflow are quite different between PGF2α-ags and the EP2 agonist OMD despite the fact that they are both members of the prostanoid receptor family of agonists [13]. Since such hypotensive therapies by PGF2α-ags and EP2 agonist medications are usually used for longer periods, knowledge of the unfavorable side effects in ocular and peri-ocular tissues would be highly desirable. In fact, deepening of the upper eyelid sulcus (DUES), a non-negligible cosmetic side effect, has been reported among long-term users of PGF2α-ags [14,15,16]. In our previous study, we reported on the pathological contribution of PGF2α-ags toward the etiology of DUES. In that study, we established a suitable model to mimic DUES by PGF2α-ags using three-dimensional (3D) tissue cultures of 3T3-L1 cells, a commonly used preadipocyte cell line [17], and human orbital fibroblasts (HOFs) [18]. The findings in that report suggested that 3D tissue cultures may be a powerful tool for screening these drug-induced effects on peri-ocular tissues. Alternatively, in our previous pilot study, we also developed a suitable in vivo model that replicates the glaucomatous human trabecular meshwork (HTM) by a 3D drop culture method using transforming growth factor (TGF)-β2-treated HTM cells [19] and evaluated the drug efficacy of inhibitors of rho-associated coiled-coil-containing protein kinases (ROCKs) using this model. This led to the suggestion that our developed 3D spheroid culture methods may also be useful for evaluating the drug-induced effects on intraocular tissues in addition to peri-ocular tissues.

It is well-known that glucocorticoids (GCs) have potent anti-inflammatory activities and therefore are frequently used in the treatment of a variety of diseases. However, GCs may also induce a number of potentially serious side effects, and, in fact, GC therapy can cause OH and secondary open-angle glaucoma [20,21]. Nevertheless, the molecular mechanisms responsible for GC-induced ocular hypertension and impaired TM cell function remain poorly understood [22,23]. To address this issue, we performed a screening of the drug-induced effects of (1) dexamethasone (DEX) toward HTM cells and (2) PGF2α-ags and an EP2 agonist toward DEX-treated HTM cells, the effects of PGF2α and OMD on several properties of the DEX-treated 2D and 3D HTM cells, including the physical properties of 3D spheroids, the size and stiffness of and the expression of major extracellular matrix (ECM) molecules, collagen (COL) 1, 4 and 6, fibronectin (FN) and α smooth muscle actin (αSMA), and their modulators, tissue inhibitors matrix proteinase (TIMP) 1–4, matrix metalloproteinase (MMP) 2, 9 and 14, and endoplasmic reticulum (ER) stress and the unfolded protein response (UPR) related factors (2D and 3D) were investigated.

## 2. Materials and Methods

### 2.1. 2D and 3D Spheroid Cultures of Human Trabecular Meshwork (HTM) Cells

Immortalized human trabecular meshwork (HTM) cells that were obtained from Applied Biological Materials Inc. (Richmond, BC, Canada) were used in the present study. 2D and 3D cultures of the HTM cells were basically performed as described previously [19]. Briefly, 2D-cultured HTM cells were further processed into 3D spheroid cultures on a hanging droplet spheroid (3D) culture plate (# HDP1385, Sigma-Aldrich Co., St.Louis, MO, USA) over a period of 6 days. In screening for the drug efficacy of the DEX (250 ng/mL)-treated 3D HTM spheroids, 100 nM PGF2α or omidenepag (OMD) were added at Day 1, and the resulting 3D cultures were maintained by changing half volumes of the medium (14 μL) each day in each well.

### 2.2. Measurements of 2D HTM Monolayers by Transendothelial Electron Resistance (TEER) and Fluorescein Isothiocyanate (FITC)-Dextran Permeability

TEER measurements of the monolayered 2D-cultured HTM cells were performed as described previously [24] using a 12-well plate for TEER (0.4 μm pore size, diameter of 12 mm; Corning Transwell, Sigma-Aldrich Co., St.Louis, MO, USA). Briefly, at approximately 80% confluence, in the presence of 250 ng/mL DEX, 100 nM omidenepag (OMD) or PGF2α were added to the apical side of the wells (Day 1), and the resulting mixture was cultured until Day 6. At Day 6, the wells were washed twice with phosphate buffered saline (PBS), and TEER (Ω cm^2^) values were measured using an electrode (Kanto Chemical Co. Inc., Tokyo, Japan) [19]. As a control, 0 Ω cm^2^ in the absence of HTM cells was confirmed.

In terms of fluorescein isothiocyanate (FITC)–dextran permeability, a 50 μmol/L solution of FITC–dextran (Sigma-Aldrich Co., St.Louis, MO, USA) was added to the basal compartments of the culture well and the culture medium from the apical compartment was collected at 60 min for the different experimental conditions. The concentrations of the FITC–dextran were measured using a multimode plate reader (Enspire; Perkin Elmer, Waltham, MA, USA) at an excitation wavelength of 490 and an emission wavelength of 530 nm. The fluorescence intensity of the control medium was used as the background concentration.

### 2.3. Measurement of the Physical Properties, Size and Solidity of 3D Spheroids

As described previously, the 3D spheroid configuration was observed by phase contrast (PC, Nikon ECLIPSE TS2; Tokyo, Japan) and the mean size of each 3D spheroid defined as the largest cross-sectional area (CSA) was determined using the Image-J software version 1.51n (National Institutes of Health, Bethesda, MD, USA) [19,25]. The physical stiffness of the 3D HTM spheroids was measured using a micro-squeezer (MicroSquisher, CellScale, Waterloo, ON, Canada) as previously reported [19,25]. Briefly, the force (μN/μm) required to compress a single spheroid on a 3-mm square plate to a 50% deformation during a 20 s interval was measured.

### 2.4. Quantitative PCR

Total RNA extraction, reverse transcription, and subsequent real-time PCR with the Universal Taqman Master mix using a StepOnePlus instrument (Applied Biosystems/Thermo Fisher Scientific) were performed as describe previously [19]. The respective cDNA values are shown as fold-change relative to the control of the normalized housekeeping gene 36B4 (*Rplp0*). Sequences of primers and Taqman probes used are listed in Supplemental Table 1.

### 2.5. Statistical Analysis

Based on statistical analyses using the Graph Pad Prism 8 (GraphPad Software, San Diego, CA, USA), the statistical significance with a confidence level greater than 95% by a two-tailed Student’s *t*-test or two-way analysis of variance (ANOVA), followed by a Tukey’s multiple comparison test, was performed as described previously [19].

## 3. Results

### 3.1. Drug- Induced Effects of PGF2α and the SELECTIVE EP2 Agonist, Omidenepag (OMD), on Monolayers of DEX-Treated HTM Cells

In screening the drug effects of PGF2α and the selective EP2 agonist, omidenepag (OMD), on DEX-treated HTM cells that are known to mimic steroid-induced glaucomatous TM, the barrier function and permeability of the 2D-cultured HTM cell monolayers were evaluated by transendothelial electron resistance (TEER) and fluorescein isothiocyanate (FITC)–dextran permeability measurements. As shown in Figure 1, the TEER values and FITC–dextran permeability were substantially increased and decreased, respectively, upon exposure to 250 ng/mL DEX. These DEX-induced effects were significantly suppressed by PGF2α or OMD, and such suppressive effects were more evident in the case of OMD. Thus, this result indicated that such suppressive effects on the DEX-induced 2D-cultured HTM monolayers may be FP- and EP2-dependent, but the effects caused by the EP2 agonist were greater that those caused by the FP agonist.

### 3.2. Drug-Induced Effects of PGF2α and the Selective EP2 Agonist, Omidenepag (OMD), on the Physical Properties, Size and Stiffness of the 3D DEX-Treated HTM Spheroids

We next evaluated the effects of PGF2α and OMD on the physical properties, size, and stiffness of the DEX-treated 3D HTM spheroids. As shown in Figure 2, during the 6-day 3D culture, the mean sizes became smaller upon the administration of 250 nM DEX at Days 3 and 6, but both PGF2α and OMD showed no significant effects on their sizes (Figure 2). However, in contrast, the physical stiffness of the 3D HTM spheroids were significantly increased by PGF2α or OMD, and these effects were more evident in the case of PGF2α (Figure 3). These results indicated that both FP and EP2 agonists may alter the physical stiffness of the 3D HTM spheroids, and that these effects were predominant in the case of the FP agonist.

### 3.3. Drug-Induced Effects of PGF2α and the Selective EP2 Agonist, Omidenepag (OMD), on Gene Expressions of ECM and Their Regulator of the 2D and 3D DEX-Treated HTM Cells

To elucidate the underlying mechanisms responsible for causing the above PGF2α and OMD effects, the mRNA expression of ECM molecules including *COL1, 4*, and *6, FN*, and α*SMA* were investigated (Figure 4). Upon administering a 250 ng/mL solution of DEX, the expression of *COL4* and α*SMA* (2D), or *FN* (3D) were significantly upregulated. The addition of PGF2α induced a significant downregulation in *COL4* and α*SMA* (2D), and *FN* (3D), and the addition of OMD induced a substantial downregulation of *COL4* (2D) and the downregulation of *COL6* (3D) and *FN* (3D). To study this issue further, qPCR analyses of *TIMP1–4*, and *MMP2, 9* and *14*, (2D and 3D) were performed. As shown in Figure 5 and Figure 6, *TIMP 4* (2D and 3D) and *MMP2 14* (3D) were significantly upregulated and downregulated, respectively, upon the administration of a 250 ng/mL solution of DEX. The addition of PGF2α or OMD induced a significant downregulation of *TIMP3* and *4* (3D) and *MMP9* (3D) and an upregulation of *MMP9* (2D), or the downregulation of *TIMP3* and *4* (3D) and *MMP9* (3D), the upregulation of *TIMP4* (2D) and *MMP2* and *14* (2D), respectively.

### 3.4. Drug-Induced Effects of PGF2α and the Selective EP2 agonist, Omidenepag (OMD), on the Gene Expression of ER Stress-Related Genes of 2D and 3D DEX-Treated HTM Cells

A substantial body of evidence exists to indicate that ER-mediated apoptosis is involved in the development and progression of several diseases including ocular diseases [26], and in fact, it has been suggested that ER stress and the UPR signaling pathway are directly implicated in the pathogenesis of glaucoma [27,28,29,30]. We therefore also examined the effects of PGF2α and the selective EP2 agonist, omidenepag (OMD), on ER stress and the UPR signaling pathway in the 2D and 3D DEX-treated HTM cells by screening ER stress-related genes corresponding to three master regulators: protein kinase RNA-like endoplasmic reticulum kinase (PERK), activating transcription factor 6 (ATF6), and inositol-requiring enzyme 1 (IRE1), and their downstream factors including glucose regulator protein (GRP)78, GRP94, and CCAAT/enhancer-binding protein homologous protein (CHOP). As shown in Figure 7, significant downregulations in IRE and ATF6, and GRP78 were observed in the DEX-treated 2D and 3D HTM cells, respectively, as compared with those of DEX-untreated 2D and 3D HTM cells. In the case of the PGF2α and OMD-induced effects, a downregulation of CHOP by OMD (2D) was only detected.

## 4. Discussion

A number of 3D culture methods have been developed recently and include scaffold-assisted techniques such as gel matrix and micro-carriers, or liquid cultures on low-attachment plates, in hanging drops or in rotation [31,32]. Although these 3D culture methods have many advantages compared to conventional 2D cultures, there are practical problems associated with each method [31,32,33,34,35,36]. Such 3D cell cultures have recently received great attention for use as suitable in vivo models for several diseases, including steroid-induced glaucoma [37,38]. In fact, it was reported that the 3D-cultured HTM cells are much more sensitive to intracellular reactive oxidative species that are induced by a hydrogen peroxide treatment compared to 2D-cultured HTM cells [39]. Furthermore, Kalouche et al. also established a three-dimensional (3D) TM cell–populated collagen gel (CPCG) model [40] and reported different drug efficacies between FP-ags and an EP2 agonist that involved a latanoprost increase and decrease in TGF-β2-mediated cell contraction and in collagen deposition, respectively, whereas butaprost inhibited both TGF-β2-dependent contraction and collagen deposition. Thus, these findings suggest that 3D culture methods may also be useful for screening drug-induced efficacies in addition to replicating in vivo disease models. In fact, we recently developed a 3D cell drop culture method as an in vivo model for Graves’ orbitopathy [25], deepening of the upper eyelid sulcus (DUES) [17,18], and as a POAG TM model using TGF-β2-treated HTM cells [19], and we simultaneously recognized that those models can be used for screening drug efficacies.

In terms of the possible mechanisms responsible for causing the hypotensive effects of both PGF2α-ags and EP2 agonists, it was suggested that uveoscleral aqueous outflow may be enhanced by these agonists, leading to an induced ECM remodeling [41,42,43,44,45]. However, since FP and EP2 receptors are also both expressed within TM tissues, it was suggested that both PGF2α-ags and EP2 agonists may also affect the conventional TM aqueous outflow [46,47,48,49]. Functionally, FP and EP2 receptors are recognized as being responsible for the contraction and relaxation, respectively, of the smooth muscle cells [50,51,52,53,54]. In fact, stimulation of the EP2 receptor causes the relaxation of Schlemm’s canal endothelial cells [54,55]. Interestingly, FP receptors have also been related to the progression of pulmonary fibrosis [56,57] and myocardial fibrosis [58]. These collective observations indicate that FP or EP2 receptors may be involved in the modulation of the structure and function of the TM in different manners. In the current study, we also found that the PF agonist, PGF2α, and the EP2 agonist, OMD, exert different effects on 2D- and 3D-cultured DEX-treated HTM cells.

ECM not only serves as a structural support within organs, but also has specific functions such as cell-to-cell signaling as well as the regulation of a variety of cellular functions [59]. COL1, 4, and 6 are major ECM components of the basement membrane (BM) [60,61,62,63], and FN is also an important ECM molecule that functions to define cell shape and contractility by association with COL1 [64]. It is known that increasing the levels of these ECM molecules reduces the ease of aqueous outflow through the TM, thus resulting in an IOP elevation [65]. It was shown that both the FP and EP2 agonists inhibited TGF-β2-mediated collagen deposition [40]. In fact, and interestingly, a decrease in collagen deposits within the TM were observed upon the administration of either latanoprost [42] or butaprost [41] to cynomolgus monkeys over a period of one year. Furthermore, another FP agonist, fluprostenol, was also reported to induce a decrease in the expression of COL4 and 6 in connective tissue growth factor (CTGF)-treated TM cells [66]. In the current study, we found that PGF2α or OMD altered the mRNA expression of some ECMs and their modulator, TIPMs and MMPs in different manners, and these alterations were also observed in both 2D and 3D HTM cultures.

In terms of POAG or steroid-induced glaucoma, increased levels of ER stress and UPR following elevated IOPs have been reported [29]. However, the issue of which steroid or elevated IOP induced such an increase in ER stress and UPR is not known at present. In the present study, upon DEX treatment some ER stress-related genes were significantly downregulated, but this regulation was different in 2D versus 3D cell cultures. Thus, the findings reported in this study suggest that elevated IOPs and not a steroid itself may rationally be related to the increase in ER stress and UPR. In fact, in support of this speculation, a previous study reported that GRP78 in human glaucomatous TM cells was downregulated [67].

The underlying mechanisms causing such a diversity in the gene expressions between 2D and 3D cell cultures remain to be elucidated. However, in our previous studies, we also detected several differences in terms of the mRNA expression of ECM molecules and other genes and in the immunolabeling of the ECMs of 3T3-L1 cells [17,68,69], human orbital fibroblasts [18], and HTM [19]. Therefore, based upon these collective findings, we speculate that the 3D spheroids may reflect different aspects of cellular structure and functions as compared to the 2D cell cultures. In addition to this, we also found that (1) the trypsin digestion of the 3D 3T3-L1 spheroids required more than 6–12 h, whereas for the 2D 3T3-L1 cells, the process was complete within a few minutes [17], and (2) the efficacies of adipogenesis of the 3D 3T3-L1 spheroids were much higher than those of 2D-cultured 3T3-L1 cells [68,69]. Therefore, the use of a combination of 2D and 3D cell cultures may be informative for the screening of drug-induced effects in addition to investigations of cellular structure and functions.

However, the current study has several limitations that need to be discussed. First, the present study was conducted using commercially available immortalized HTM cells that were certified as being true HTM cells and not primary cultured HTM cells. Since, in terms of HTM cells, significant biological variabilities are present from donor to donor, using one HTM cell line in this study might have been insufficient to determine whether the obtained effects were representative across donor tissues/cells. Nevertheless, our national laws do not permit the use of human donor eyes for research purposes. Given this situation, we had no choice other than to use these commercially available immortalized HTM cells. However, the immortalized HTM cells used in this study clearly had the advantage of reproducibility compared to the primary cultured HTM cells, which vary with advancing passaging, and therefore we believe that the former may be more suitable for the present study, the purpose of which was screening drug-induced effects. Second, it is known that in primary cultured HTM cells, tight junctions, the major components that contribute to TEER, generally are not present, although Schlemm’s canal cells show the presence of tight junction proteins, such as claudin-1 and 5 [70]. In terms of the mechanisms responsible for causing the TEER changes after the administration of PGFα or OMD in 2D immortalized HTM cells, we assumed that these drug-induced effects in ECM deposits that were enhanced by TGF-β2 may be involved, as was observed in our previous study by scanning electron microscopy [19]. Third, in the present study, we used quantitative real-time RT-PCR to monitor the changes in the mRNA expression of several genes of interest in 2D as well as 3D cultured HTM cells under different treatment conditions, since previous studies demonstrated almost similar characteristics that were compared based on mRNA expression and immunolabeling of the target molecules [9,17,25,68,69]. However, mRNA expression may not correlate with protein expression due to posttranslational modification, and some genes may have alternatively spliced isoforms, and the function of a gene is generally based on its protein product. Fourth, our ultimate study goal was to elucidate drug-induced effects on the structure and function of the human TM under aqueous humor (AH) circulation by using suitable in vivo models. Therefore, to accomplish this, the current investigation involved the use of 2D HTM monolayers as well as 3D HTM spheroids, and some interesting results were obtained, as described above. However, our 2D and 3D cultured model may still be insufficient for accurately evaluating AH circulation-related effects. Therefore, an additional study using primary cultured HTM cells obtained from donors with or without glaucoma, immunoelectron microscopy to study ECM deposits, and millifluidic technologies [71] to evaluate circulatory dynamics should be considered.

## Figures and Tables

**Figure 1 biomedicines-09-00930-f001:**
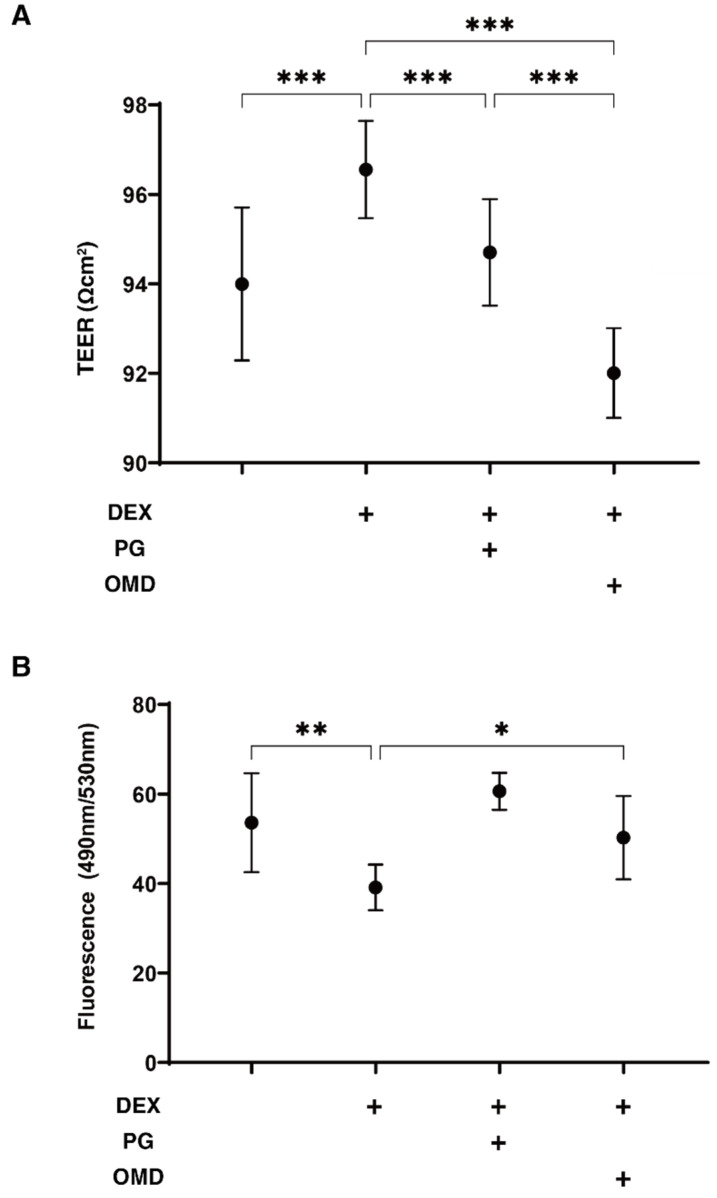
Effects of PGF2α and the EP2 agonist, omidenepag (OMD), on transendothelial electrical resistance (TEER) (**A**) and FITC–dextran permeability (**B**) of DEX-treated 2D culture of HTM cell monolayers. To evaluate the effects of 100 nM PGF2α or EP2 agonist, omidenepag (OMD), on barrier function (Ω cm^2^) and the permeability of DEX-untreated or treated 2D-cultured HTM monolayers, TEER (panel A) and FITC–dextran permeability (panel B) measurements were performed, respectively. All experiments were performed in triplicate using fresh preparations (*n* = 4). Data are presented as the arithmetic mean ± standard error of the mean (SEM). * *p* < 0.05 ** *p* < 0.01,*** *p* < 0.005 (ANOVA followed by a Tukey’s multiple comparison test).

**Figure 2 biomedicines-09-00930-f002:**
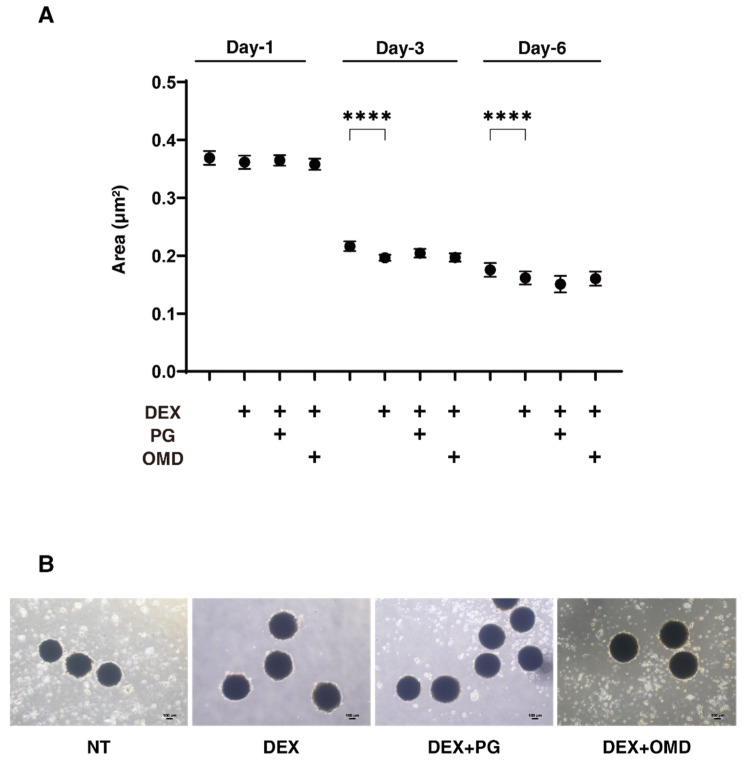
Changes in the sizes of DEX-treated 3D HTM spheroids at Days 1, 3 or 6 in the presence and absence of PGF2α or the EP2 agonist, omidenepag (OMD). At Day 1, 3 or 6, the mean sizes of HTM 3D spheroids and those treated by 250 ng/mL DEX were plotted in the absence and presence of 100 nM PGF2α or EP2 agonist, omidenepag (OMD) (panel **A**). Representative phase contrast microscopic images at Day 6 are shown in (panel **B**). Data are presented as the arithmetic mean ± standard error of the mean (SEM). **** *p* < 0.001 (ANOVA followed by a Tukey’s multiple comparison test).

**Figure 3 biomedicines-09-00930-f003:**
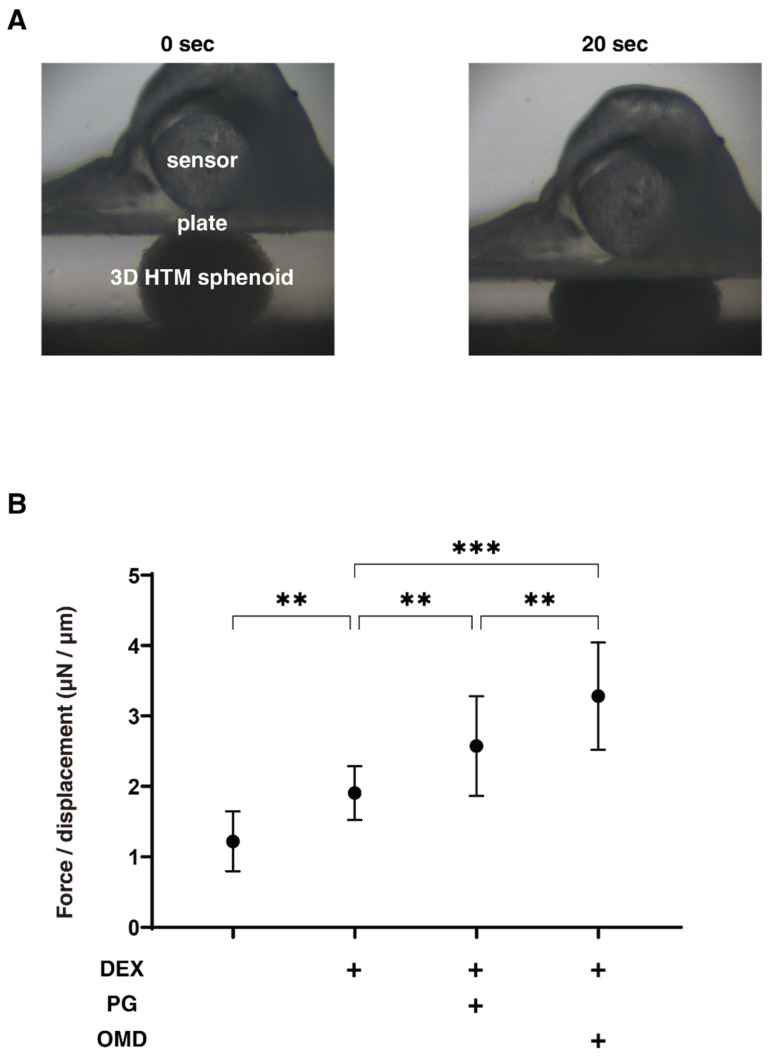
Physical solidity of DEX-treated 3D HTM spheroids at Day 6 in the presence and absence of PGF2α or the EP2 agonist, omidenepag (OMD). At Day 6, the physical solidity of a single 3D HTM spheroid was measured by a micro-squeezer (panel (**A**)). The force requiring the 50% deformity during 20 s (μN/μm force/displacement) of HTM 3D spheroids (control) and those treated with a 250 ng/mL solution of DEX in the absence and presence of 100 nM PGF2α or EP2 agonist, omidenepag (OMD), are plotted in panel (**B**). ** *p* < 0.01, *** *p* < 0.005 (ANOVA followed by a Tukey’s multiple comparison test).

**Figure 4 biomedicines-09-00930-f004:**
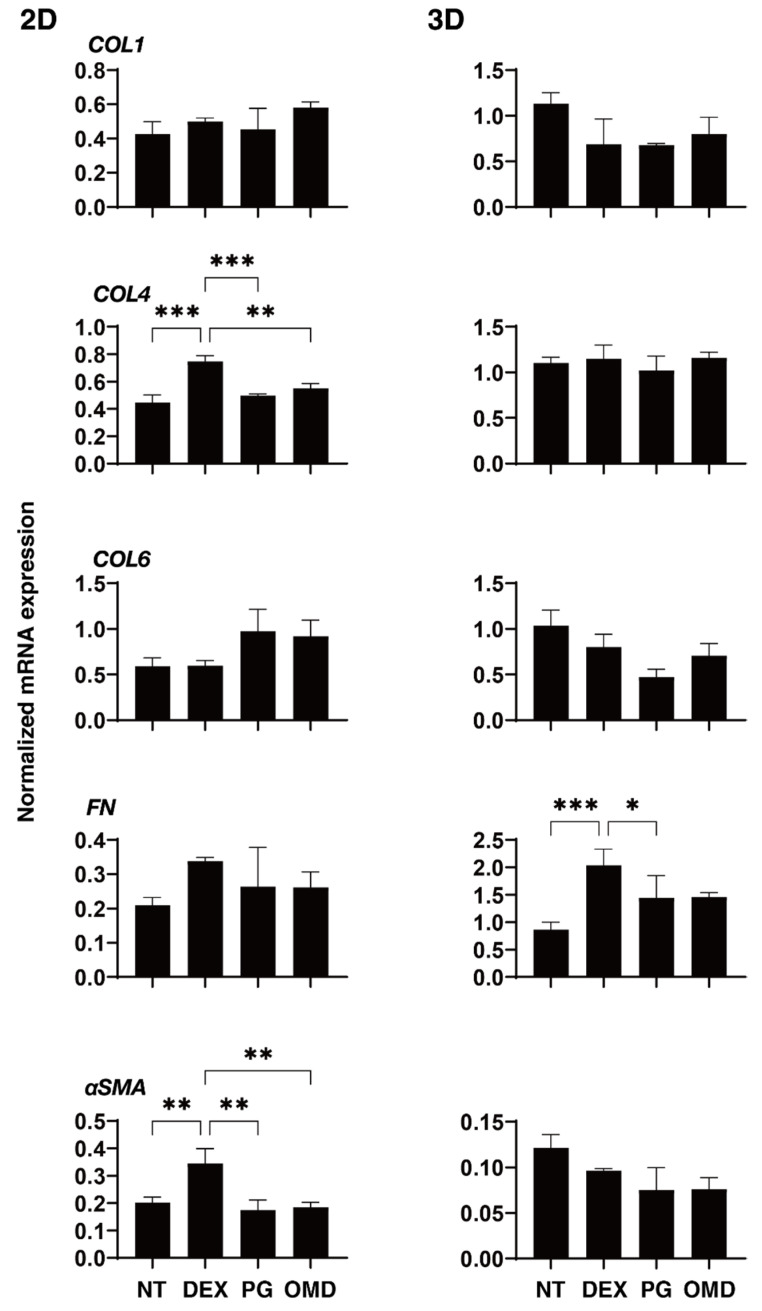
mRNA expression of ECM in DEX-treated 2D and 3D HTM cells at Day 6 in the presence and absence of PGF2α or the EP2 agonist, omidenepag (OMD). At Day 6, HTM 2D cells and 3D spheroids and those treated with a 250 ng/mL solution of DEX in the absence and presence of 100 nM PGF2α or EP2 agonist, omidenepag (OMD), were subjected to qPCR analysis to estimate the expression of mRNA in ECMs (*COL1*, *COL4*, *COL6*, *FN* and *aSMA*). All experiments were performed in duplicate using fresh preparations. Data are presented as the arithmetic mean ± standard error of the mean (SEM). * *p* < 0.05, ** *p* < 0.01, *** *p* < 0.005 (ANOVA followed by a Tukey’s multiple comparison test).

**Figure 5 biomedicines-09-00930-f005:**
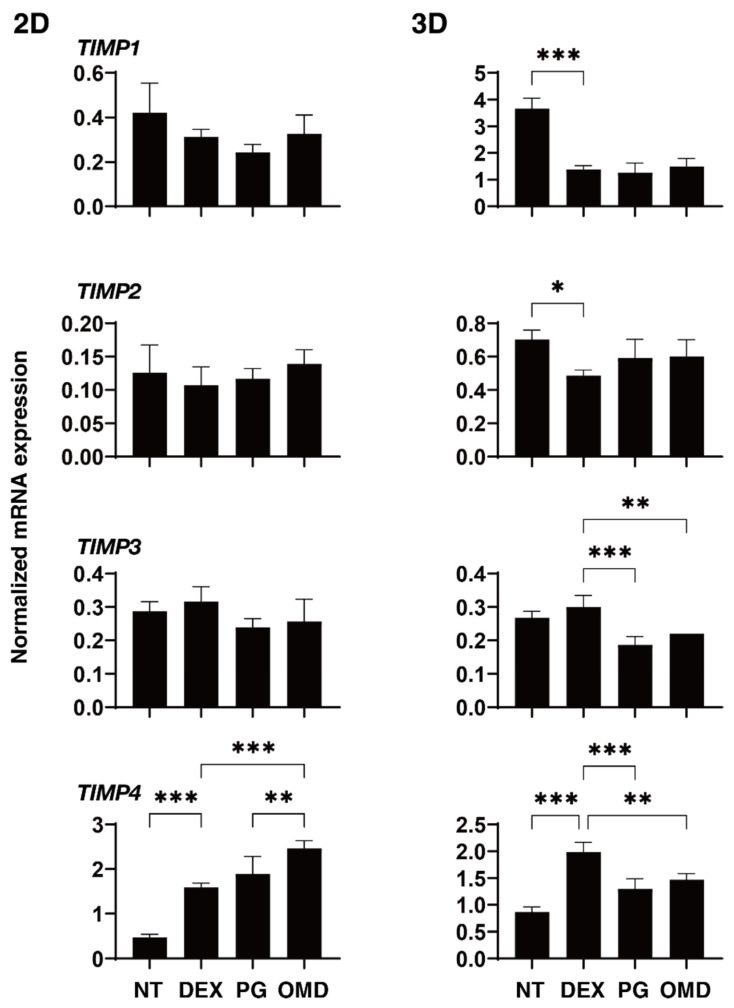
mRNA expression of TIMPs in DEX-treated 2D and 3D HTM cells at Day 6 in the presence and absence of PGF2α or the EP2 agonist, omidenepag (OMD). At Day 6, HTM 2D cells and 3D spheroids and those treated with a 250 ng/mL solution of DEX in the absence and presence of 100 nM PGF2α or EP2 agonist, omidenepag (OMD), were subjected to qPCR analysis to estimate the expression of mRNA in *TIMP1–4*. All experiments were performed in duplicate using fresh preparations. Data are presented as the arithmetic mean ± standard error of the mean (SEM). * *p* < 0.05, ** *p* < 0.01, *** *p* < 0.005 (ANOVA followed by a Tukey’s multiple comparison test).

**Figure 6 biomedicines-09-00930-f006:**
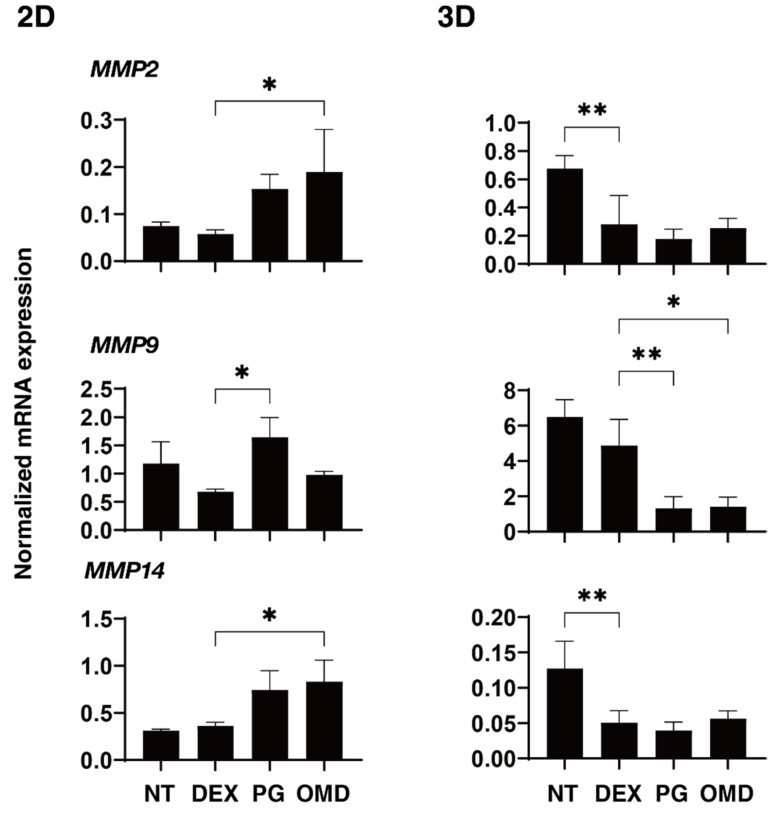
mRNA expression of MMPs and CTGF in DEX-treated 2D and 3D HTM cells at Day 6 in the presence and absence of PGF2α or the EP2 agonist, omidenepag (OMD). At Day 6, HTM 2D cells and 3D spheroids and those treated with a 250 ng/mL solution of DEX in the absence and presence of 100 nM PGF2α or EP2 agonist, omidenepag (OMD), were subjected to qPCR analysis to estimate the expression of mRNA in *MMP 2, 9* and *14* and *CTGF*. All experiments were performed in duplicate using fresh preparations. Data are presented as the arithmetic mean ± standard error of the mean (SEM). * *p* < 0.05, ** *p* < 0.01, (ANOVA followed by a Tukey’s multiple comparison test).

**Figure 7 biomedicines-09-00930-f007:**
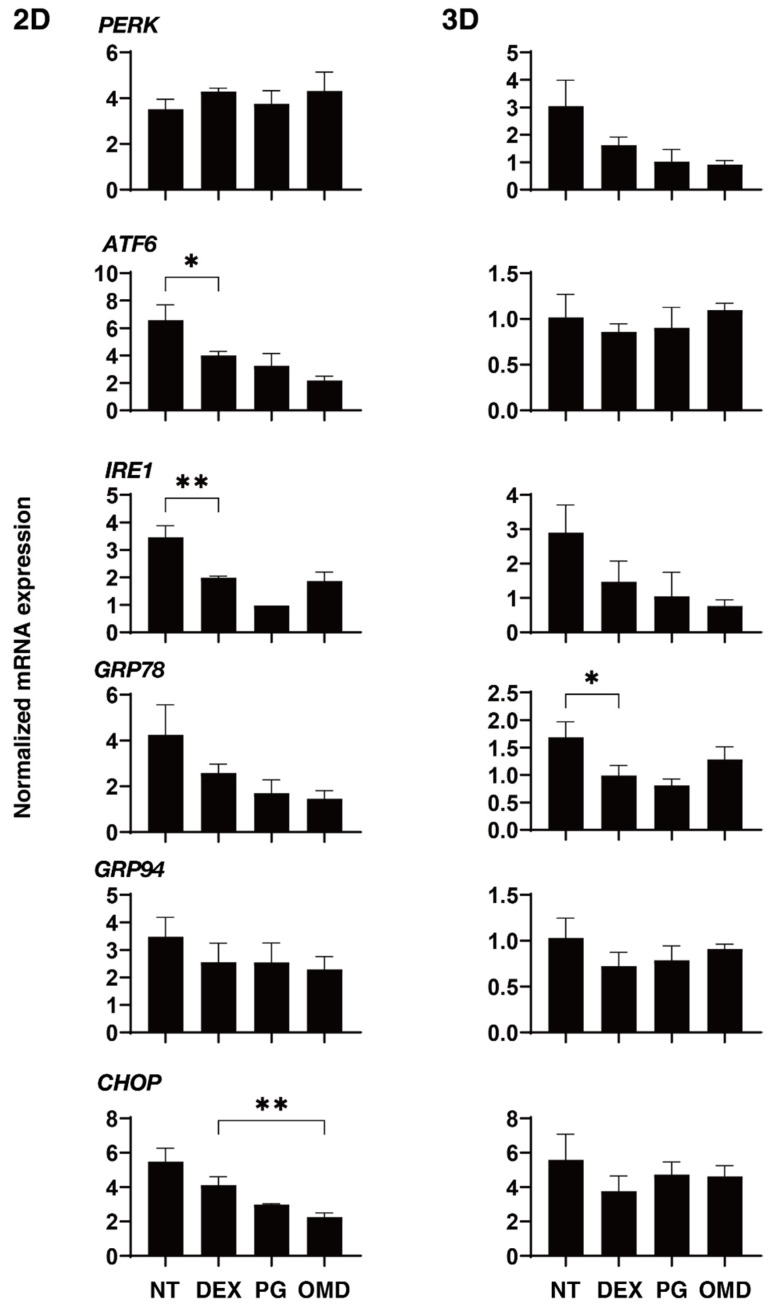
mRNA expression of major ER stress-related genes in DEX-treated 2D and 3D HTM cells at Day 6 in the presence and absence of PGF2α or the EP2 agonist, omidenepag (OMD). At Day 6, HTM 2D cells and 3D spheroids and those treated with a 250 ng/mL solution of DEX in the absence and presence of 100 nM PGF2α or EP2 agonist, omidenepag (OMD), were subjected to qPCR analysis to estimate the expression of mRNA in major ER stress-related genes of three master regulators: protein kinase RNA-like endoplasmic reticulum kinase (PERK), activating transcription factor 6 (ATF6), and inositol-requiring enzyme 1 (IRE1), and their downstream factors including glucose regulator protein (GRP)78, GRP94, and CCAAT/enhancer-binding protein homologous protein (CHOP). All experiments were performed in duplicate using fresh preparations. Data are presented as the arithmetic mean ± standard error of the mean (SEM). * *p* < 0.05, ** *p* < 0.01 (ANOVA followed by a Tukey’s multiple comparison test).

**Table 1 biomedicines-09-00930-t001:** Sequences of primers used in the qPCR.

Sequence			Exon Location	RefSeq Number
human RPLP0	Probe Primer2 Primer1	5′-/56-FAM/CCCTGTCTT/ZEN/CCCTGGGCATCAC/3IABkFQ/-3′ 5′-TCGTCTTTAAACCCTGCGTG-3′ 5′-TGTCTGCTCCCACAATGAAAC-3′	2-3	NM_001002
human COL1A1	Probe Primer2 Primer1	5′-/56-FAM/TCGAGGGCC/ZEN/AAGACGAAGACATC/3IABkFQ/-3′ 5′-GACATGTTCAGCTTTGTGGAC-3′ 5′-TTCTGTACGCAGGTGATTGG-3′	1-2	NM_000088
human COL6A1	Primer2 Primer1	5′-CCTCGTGGACAAAGTCAAGT-3′ 5′-GTGAGGCCTTGGATGATCTC-3′	2-3	NM_001848
human FN1	Primer2 Primer1	5′-CGTCCTAAAGACTCCATGATCTG-3′ 5′-ACCAATCTTGTAGGACTGACC-3′	3-4	NM_212482
human αSMA	Probe Primer2 Primer1	5′-/56-FAM/AGACCCTGT/ZEN/TCCAGCCATCCTTC/3IABkFQ/-3′ 5′-AGAGTTACGAGTTGCCTGATG-3′ 5′-CTGTTGTAGGTGGTTTCATGGA-3′	8-9	NM_001613
human Perk	Forward Reverse	5′-ACGATGAGACAGAGTTGCGAC-3′ 5′-AATCCCACTGCTTTTTACCATGA-3′		NM_004836
human Atf6a	Forward Reverse	5′-TCAGACAGTACCAACGCTTATGC-3′ 5′-GTTGTACCACAGTAGGCTGAGA-3		NM_007348
human Ire1a	Forward Reverse	5′-TTTGGAAGTACCAGCACAGTG-3′ 5′-TGCCATCATTAGGATCTGGGA-3′		NM_001433
human Grp78	Forward Reverse	5′-CATCACGCCGTCCTATGTCG-3′ 5′-CGTCAAAGACCGTGTTCTCG-3′		NM_005347
human Grp94	Forward Reverse	5′-CTGGGACTGGGAACTTATGAATG-3′ 5′-TCCATATTCGTCAAACAGACCAC-3′		NM_003299
human Chop	Forward Reverse	5′-GGAGAACCAGGAAACGGAAAC-3′ 5′-TCTCCTTCATGCGCTGCTTT-3′		NM_004083

## Data Availability

The data that support the findings of this study are available from the corresponding author upon reasonable request.

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
