# Peer review of "Screening of the Drug-Induced Effects of Prostaglandin EP2 and FP Agonists on 3D Cultures of Dexamethasone-Treated Human Trabecular Meshwork Cells"

_biomedicines, 2021, doi:10.3390/biomedicines9080930_

Round 1

Reviewer 1 Report

This manuscript by Watanabe et al. report the finding of the prostaglandin EP2 and FP agonists on 3D cultured DEX treated human trabecular meshwork (HTM) cells formed sphenoids. This is an original research report by using a variety of techniques with solid experimental evidence. I have some questions or concerns that need to be addressed.

  1. The authors used immortalized HTM cells in their study. Is there any advantage or disadvantage compared with using primary cultured HTM cells? Please give a brief discussion.

  1. The authors used 100 µM PGFα or OMD in the experiments, if the 100 µM concentration has been used before or their half-lives are known, please cite it.

  1. The authors showed the presence of relatively strong TEER (around 90ê­¥cm2) in 2D cultured HTM cells. Did the authors also record the TEER in 12 well plates that only contain cell culture medium (no HTM cells in the wells) , this can be used as a control to show the TEER reading is close to “0 ê­¥cm2” .  In primary cultured HTM cells, tight junctions, the major components that contribute to TEER, generally are not present. However, the Schlemm’s canal cells show the presence of tight junction proteins, such as claudin-1 and 5.  Please give some discussion regarding the mechanism that cause TEER changes after administration of PGFα or OMD in 2D immortalized HTM cells.

  1. The authors also used quantitative real time RT-PCR to monitor the changes in mRNA expression of many interested genes in 2D, as well as 3D cultured HTM cells under different treatment conditions. One issue is the mRNA expression may not correlated with protein expression due to posttranslational modification, as well as some genes may have alternatively spliced isoforms, and the function of a gene is generally completed via its protein product. Please give some discussion regarding the gene expression and protein function.

  1. In page 2, 2.1. section, please change “D and 3D sphenoid cultures …” to “2D and 3D sphenoid cultures …”.

  1. In the last paragraph at page 10, please delete “ This section may be divided by subheadings. It should provide a concise and precise description of the experimental results, their interpretation, as well as the experimental conclusions that can be drawn”.

Thank you for the invitation!

Author Response

Dear Editor,

Thank you very much for the constructive comments concerning our manuscript; " Screening of the drug induced effects of prostaglandin EP2 and FP agonists on 3D cultures of dexamethasone-treated human trabecular meshwork cells“. We carefully examined all of the comments from the Reviewer and have made a series of specific changes to our manuscript as follows;

Reviewer1

This manuscript by Watanabe et al. report the finding of the prostaglandin EP2 and FP agonists on 3D cultured DEX treated human trabecular meshwork (HTM) cells formed spheroids. This is an original research report by using a variety of techniques with solid experimental evidence. I have some questions or concerns that need to be addressed.

  1. The authors used immortalized HTM cells in their study. Is there any advantage or disadvantage compared with using primary cultured HTM cells? Please give a brief discussion.

Answer; Thank you for this comment. As suggested, brief discussion related advantage or disadvantage compared with using primary cultured HTM cells is included as study limitation in the last paragraph of discussion; “However, the current study has several limitations that need to be discussed; First, the present study was conducted using commercially available immortalized HTM cells which were certified as being true HTM cells, and not primary cultured HTM cells. Since, in terms of HTM cells, significant biological variabilities are present from donor to donor, using one HTM cell line in this study might have been insufficient to determine whether the obtained effects are representative across donor tissues/cells. Nevertheless, our national laws do not permit the use of human donor eyes for research purposes. Given this situation, we have no choice other than to use these commercially available immortalized HTM cells. However, the immortalized HTM cells used in this study clearly have the advantage of reproducibility compared to the primary cultured HTM cells, which vary with advancing passaging, and therefore we believe that the former may be more suitable for the present study, the purpose of was screening drug induced effects. Second, it is known that in primary cultured HTM cells, tight junctions, the major components that contribute to TEER, generally are not present, although Schlemm’s canal cells show the presence of tight junction proteins, such as claudin-1 and 5 [70]. In terms of the mechanisms responsible for causing the TEER changes after the administration of PGFα or OMD in 2D immortalized HTM cells, we assumed that these drugs induced effects toward ECM deposits that were enhanced by TGF-b2 may be involved, as was observed in our previous study by scanning electron microscopy [19]. Third, in the present study, we used quantitative real time RT-PCR to monitor the changes in mRNA expression of several genes of interest in 2D, as well as 3D cultured HTM cells under different treatment conditions since previous studies demonstrated almost similar characteristics that were compared based on mRNA expression and immunolabeling of the target molecules [9,17,25,68,69]. However, mRNA expression may not correlate with protein expression due to posttranslational modification, and some genes may have alternatively spliced isoforms, and the function of a gene is generally based on its protein product. Forth, our ultimate study goal was to elucidate drug induced effects on the structure and function of the human TM under aqueous humor (AH) circulation by using suitable in vivo models. Therefore, to accomplish this, the current investigation involved the use of 2D HTM monolayers as well as 3D HTM spheroids, and some interesting results were obtained, as described above. However, our 2D and 3D cultured model may be still insufficient for accurately evaluating AH circulation related effects. Therefore, an additional study using primary cultured HTM cells obtained from donors with or without glaucoma, immunoelectron microscopy to study ECM deposits, millifluidic technologies [71] to evaluate circulatory dynamics others should be considered.”.

  1. The authors used 100 µM PGFα or OMD in the experiments, if the 100 µM concentration has been used before or their half-lives are known, please cite it.

Answer; Thank you for this critical comment related to the concentrations of PGFα or OMD, and deeply apologized our careless mistake because this 100 µM was wrong, and 100 nM should be corrected.This information was already sent to Journal office to correct them.

  1. The authors showed the presence of relatively strong TEER (around 90ê­¥cm2) in 2D cultured HTM cells. Did the authors also record the TEER in 12 well plates that only contain cell culture medium (no HTM cells in the wells), this can be used as a control to show the TEER reading is close to “0 ê­¥cm2” .  In primary cultured HTM cells, tight junctions, the major components that contribute to TEER, generally are not present. However, the Schlemm’s canal cells show the presence of tight junction proteins, such as claudin-1 and 5.  Please give some discussion regarding the mechanism that cause TEER changes after administration of PGFα or OMD in 2D immortalized HTM cells.

Answer; As suggested, we already confirmed that the TEER reading is close to “0  ê­¥cm2” in the control without HTM cells. This information is included in the method; “TEER measurements of the monolayered 2D cultured HTM cells were performed as described previously [24] using a 12 well plate for TEER (0.4 μm pore size, diameter of 12 mm; Corning Transwell, Sigma-Aldrich). Briefly, at approximately 80 % confluence, in the presence of 250 ng/mL DEX, 100 nM Omidenepag (OMD) or PGF2a were added the apical side of the wells (Day 1), and the resulting mixture was cultured until Day 6. At Day 6, the wells were washed twice with phosphate buffered saline (PBS), and TEER (Ωcm2) the values were measured using an electrode (KANTO CHEMICAL CO. INC., Tokyo, Japan) [19]. As a control, 0 Ωcm2 in the absence of HTM cells was confirmed.”. In addition, the possible mechanism causing TEER value changes by drugs is also included in the study limitation in the last paragraph of Discussion; “However, the current study has several limitations that need to be discussed; First, the present study was conducted using commercially available immortalized HTM cells which were certified as being true HTM cells, and not primary cultured HTM cells. Since, in terms of HTM cells, significant biological variabilities are present from donor to donor, using one HTM cell line in this study might have been insufficient to determine whether the obtained effects are representative across donor tissues/cells. Nevertheless, our national laws do not permit the use of human donor eyes for research purposes. Given this situation, we have no choice other than to use these commercially available immortalized HTM cells. However, the immortalized HTM cells used in this study clearly have the advantage of reproducibility compared to the primary cultured HTM cells, which vary with advancing passaging, and therefore we believe that the former may be more suitable for the present study, the purpose of was screening drug induced effects. Second, it is known that in primary cultured HTM cells, tight junctions, the major components that contribute to TEER, generally are not present, although Schlemm’s canal cells show the presence of tight junction proteins, such as claudin-1 and 5 [70]. In terms of the mechanisms responsible for causing the TEER changes after the administration of PGFα or OMD in 2D immortalized HTM cells, we assumed that these drugs induced effects toward ECM deposits that were enhanced by TGF-b2 may be involved, as was observed in our previous study by scanning electron microscopy [19]. Third, in the present study, we used quantitative real time RT-PCR to monitor the changes in mRNA expression of several genes of interest in 2D, as well as 3D cultured HTM cells under different treatment conditions since previous studies demonstrated almost similar characteristics that were compared based on mRNA expression and immunolabeling of the target molecules [9,17,25,68,69]. However, mRNA expression may not correlate with protein expression due to posttranslational modification, and some genes may have alternatively spliced isoforms, and the function of a gene is generally based on its protein product. Forth, our ultimate study goal was to elucidate drug induced effects on the structure and function of the human TM under aqueous humor (AH) circulation by using suitable in vivo models. Therefore, to accomplish this, the current investigation involved the use of 2D HTM monolayers as well as 3D HTM spheroids, and some interesting results were obtained, as described above. However, our 2D and 3D cultured model may be still insufficient for accurately evaluating AH circulation related effects. Therefore, an additional study using primary cultured HTM cells obtained from donors with or without glaucoma, immunoelectron microscopy to study ECM deposits, millifluidic technologies [71] to evaluate circulatory dynamics others should be considered.”.

  1. The authors also used quantitative real time RT-PCR to monitor the changes in mRNA expression of many interested genes in 2D, as well as 3D cultured HTM cells under different treatment conditions. One issue is the mRNA expression may not correlated with protein expression due to posttranslational modification, as well as some genes may have alternatively spliced isoforms, and the function of a gene is generally completed via its protein product. Please give some discussion regarding the gene expression and protein function.

Answer; As suggested, a discussion of the possibilities is now included in the study limitation of the last paragraph of Discussion; “However, the current study has several limitations that need to be discussed; First, the present study was conducted using commercially available immortalized HTM cells which were certified as being true HTM cells, and not primary cultured HTM cells. Since, in terms of HTM cells, significant biological variabilities are present from donor to donor, using one HTM cell line in this study might have been insufficient to determine whether the obtained effects are representative across donor tissues/cells. Nevertheless, our national laws do not permit the use of human donor eyes for research purposes. Given this situation, we have no choice other than to use these commercially available immortalized HTM cells. However, the immortalized HTM cells used in this study clearly have the advantage of reproducibility compared to the primary cultured HTM cells, which vary with advancing passaging, and therefore we believe that the former may be more suitable for the present study, the purpose of was screening drug induced effects. Second, it is known that in primary cultured HTM cells, tight junctions, the major components that contribute to TEER, generally are not present, although Schlemm’s canal cells show the presence of tight junction proteins, such as claudin-1 and 5 [70]. In terms of the mechanisms responsible for causing the TEER changes after the administration of PGFα or OMD in 2D immortalized HTM cells, we assumed that these drugs induced effects toward ECM deposits that were enhanced by TGF-b2 may be involved, as was observed in our previous study by scanning electron microscopy [19]. Third, in the present study, we used quantitative real time RT-PCR to monitor the changes in mRNA expression of several genes of interest in 2D, as well as 3D cultured HTM cells under different treatment conditions since previous studies demonstrated almost similar characteristics that were compared based on mRNA expression and immunolabeling of the target molecules [9,17,25,68,69]. However, mRNA expression may not correlate with protein expression due to posttranslational modification, and some genes may have alternatively spliced isoforms, and the function of a gene is generally based on its protein product. Forth, our ultimate study goal was to elucidate drug induced effects on the structure and function of the human TM under aqueous humor (AH) circulation by using suitable in vivo models. Therefore, to accomplish this, the current investigation involved the use of 2D HTM monolayers as well as 3D HTM spheroids, and some interesting results were obtained, as described above. However, our 2D and 3D cultured model may be still insufficient for accurately evaluating AH circulation related effects. Therefore, an additional study using primary cultured HTM cells obtained from donors with or without glaucoma, immunoelectron microscopy to study ECM deposits, millifluidic technologies [71] to evaluate circulatory dynamics others should be considered.”.

  1. In page 2, 2.1. section, please change “D and 3D spheroid cultures …” to “2D and 3D spheroid cultures …”.

Answer; Thank you for this comment. As suggested, this was changed to “2D and 3D spheroid cultures of human trabecular meshwork (HTM) Cells”

  1. In the last paragraph at page 10, please delete “This section may be divided by subheadings. It should provide a concise and precise description of the experimental results, their interpretation, as well as the experimental conclusions that can be drawn”.

Answer; As pointed out, erroneous sentences “This section may be divided by subheadings. It should provide a concise and precise description of the experimental results, their interpretation, as well as the experimental conclusions that can be drawn” was deleted.

Thank you for the invitation!

Reviewer 2 Report

In this manuscript the authors  reported an interesting approach for  analyse   the effects of selected drug ( prostaglandin F2α   and EP2 agonist, omidenapag (OMD) using two- and three-dimensional (2D and 3D) human trabecular meshwork (HTM) cultures after treatment with dexamethasone (DEX). The innovative approach is the use of spheroids as 3D model in line with the actual experimental model approach to mimic better the real in vivo conditions.

The experimental design was well outlined with advanced molecular biology approaches which lead to highlights the different results of response using 2D or 3D spheroids,

 2D HTM monolayers were analysed  by Transendothelial electron resistance (TEER) and Fluorescein isothiocyanate (FITC)-dextran permeability. The 3D sphenoid configuration was observed by phase contrast and the largest cross-sectional area was extrapolated by the mean size of each 3D sphenoid. Moreover, the physical stiffness of the 3D HTM sphenoids was measured using a micro-squeezer. MRNA techmique was applyied for analysing gene expression of  selected biomolecules of  ECM,   including collagen   1, 4 and 6, and fibronectin ,  αSMA), a tissue inhibitor of metalloproteinase (TIMP) 1-4, matrix metalloproteinase (MMP) 2, 9 and 14 and endoplasmic reticulum (ER) stress related factors,, such as IRE1,  ATF6.   (GRP)78  and CHOP. The effects of selected drugs on th 2D and spheroids  HTM in vitro models highlights that  2D model did not   show the same behaviour of 3D spheroids models. The authors have to consider also the potentiality of applying  millifluidic technologies to their spheroids since with this approach the circulation of culture medium can be consider more reliable than static cultures. 

Author Response

Dear Editor,

Thank you very much for the constructive comments concerning our manuscript; " Screening of the drug induced effects of prostaglandin EP2 and FP agonists on 3D cultures of dexamethasone-treated human trabecular meshwork cells“. We carefully examined all of the comments from the Reviewer and have made a series of specific changes to our manuscript as follows;

Reviewer2

In this manuscript the authors reported an interesting approach for analyse the effects of selected drug (prostaglandin F2α and EP2 agonist, omidenapag (OMD) using two- and three-dimensional (2D and 3D) human trabecular meshwork (HTM) cultures after treatment with dexamethasone (DEX). The innovative approach is the use of spheroids as 3D model in line with the actual experimental model approach to mimic better the real in vivo conditions.

The experimental design was well outlined with advanced molecular biology approaches which lead to highlights the different results of response using 2D or 3D spheroids,

 2D HTM monolayers were analysed by Transendothelial electron resistance (TEER) and Fluorescein isothiocyanate (FITC)-dextran permeability. The 3D spheroid configuration was observed by phase contrast and the largest cross-sectional area was extrapolated by the mean size of each 3D spheroid. Moreover, the physical stiffness of the 3D HTM spheroids was measured using a micro-squeezer. MRNA techmique was applyied for analysing gene expression of selected biomolecules of ECM, including collagen 1, 4 and 6, and fibronectin, αSMA), a tissue inhibitor of metalloproteinase (TIMP) 1-4, matrix metalloproteinase (MMP) 2, 9 and 14 and endoplasmic reticulum (ER) stress related factors, such as IRE1, ATF6, (GRP)78 and CHOP. The effects of selected drugs on th 2D and spheroids HTM in vitro models highlights that 2D model did not show the same behaviour of 3D spheroids models. The authors have to consider also the potentiality of applying millifluidic technologies to their spheroids since with this approach the circulation of culture medium can be consider more reliable than static cultures.

Answer; Thank you so much for such a constructive comment. This information is included in the study limitations in the last paragraph of Discussion; “However, the current study has several limitations that need to be discussed; First, the present study was conducted using commercially available immortalized HTM cells which were certified as being true HTM cells, and not primary cultured HTM cells. Since, in terms of HTM cells, significant biological variabilities are present from donor to donor, using one HTM cell line in this study might have been insufficient to determine whether the obtained effects are representative across donor tissues/cells. Nevertheless, our national laws do not permit the use of human donor eyes for research purposes. Given this situation, we have no choice other than to use these commercially available immortalized HTM cells. However, the immortalized HTM cells used in this study clearly have the advantage of reproducibility compared to the primary cultured HTM cells, which vary with advancing passaging, and therefore we believe that the former may be more suitable for the present study, the purpose of was screening drug induced effects. Second, it is known that in primary cultured HTM cells, tight junctions, the major components that contribute to TEER, generally are not present, although Schlemm’s canal cells show the presence of tight junction proteins, such as claudin-1 and 5 [70]. In terms of the mechanisms responsible for causing the TEER changes after the administration of PGFα or OMD in 2D immortalized HTM cells, we assumed that these drugs induced effects toward ECM deposits that were enhanced by TGF-b2 may be involved, as was observed in our previous study by scanning electron microscopy [19]. Third, in the present study, we used quantitative real time RT-PCR to monitor the changes in mRNA expression of several genes of interest in 2D, as well as 3D cultured HTM cells under different treatment conditions since previous studies demonstrated almost similar characteristics that were compared based on mRNA expression and immunolabeling of the target molecules [9,17,25,68,69]. However, mRNA expression may not correlate with protein expression due to posttranslational modification, and some genes may have alternatively spliced isoforms, and the function of a gene is generally based on its protein product. Forth, our ultimate study goal was to elucidate drug induced effects on the structure and function of the human TM under aqueous humor (AH) circulation by using suitable in vivo models. Therefore, to accomplish this, the current investigation involved the use of 2D HTM monolayers as well as 3D HTM spheroids, and some interesting results were obtained, as described above. However, our 2D and 3D cultured model may be still insufficient for accurately evaluating AH circulation related effects. Therefore, an additional study using primary cultured HTM cells obtained from donors with or without glaucoma, immunoelectron microscopy to study ECM deposits, millifluidic technologies [71] to evaluate circulatory dynamics others should be considered.”.

Round 2

Reviewer 1 Report

The authors have addressed all my concerns and suggestions. 

Thank you for the invitation!